# Trigger factor chaperone acts as a mechanical foldase

Shubhasis Haldar [1], Rafael Tapia-Rojo[1], Edward C. Eckels[1], Jessica Valle-Orero[1] & Julio M. Fernandez[1]

Proteins fold under mechanical forces in a number of biological processes, ranging from muscle contraction to co-translational folding. As force hinders the folding transition, chaperones must play a role in this scenario, although their influence on protein folding under force has not been directly monitored yet. Here, we introduce single-molecule magnetic tweezers to study the folding dynamics of protein L in presence of the prototypical molecular chaperone trigger factor over the range of physiological forces (4–10 pN). Our results show that trigger factor increases prominently the probability of folding against force and accelerates the refolding kinetics. Moreover, we find that trigger factor catalyzes the folding reaction in a force-dependent manner; as the force increases, higher concentrations of trigger factor are needed to rescue folding. We propose that chaperones such as trigger factor can work as foldases under force, a mechanism which could be of relevance for several physiological processes.

[1] Department of Biological Sciences, Columbia University, New York, NY 10027, USA. Shubhasis Haldar and Rafael Tapia-Rojo contributed equally to this work. Correspondence and requests for materials should be addressed to S.H. (email: sh3529@columbia.edu) or to R.T-R. (email: rt2605@columbia.edu) or to J.M.F. (email: jfernandez@columbia.edu)

Force is being recognized as playing an increasingly important role in many biological systems[1–7]. In particular, there is growing evidence that proteins must fold against a pulling force in a number of situations, such as the titin domains during muscle contraction[1] or co-translational folding at the mouth of the ribosome[7]. The mechanical force tilts the free energy landscape towards the unfolded state, hampering the refolding transition. In this regard, molecular chaperones—well known to assist protein folding through a variety of mechanisms[8–13]—might play a relevant role by favoring the folding transition or by allowing protein folding to occur at higher mechanical loads. Nevertheless, to date, it has not been possible to study the direct influence of chaperones on protein folding under force, mainly due to the instrumental limitations that have prevented monitoring single refolding events over long time ranges.

Trigger factor (TF) is one of the prototypical chaperones of *Escherichia coli* that exists in the cell in both ribosome-bound and free cytosolic states. Ribosomal bound TF interacts directly with the nascent polypeptide chain coming out of the ribosome and assists protein folding through different well-described mechanisms[14–19]. Notably, when the nascent chain emerges from the ribosome, a force is exerted on the chain by co-translational protein folding on the edge of the ribosome, or from electrostatic forces as stretches of charged residues are inserted through the lipid bilayer[2, 4, 5, 7]. Force transmission is crucial for resolving stalls and pauses induced by interactions of the nascent polypeptide with the ribosomal exit tunnel. The SecM arrest peptide, for example, requires picoNewton (pN) level forces to reach all the way to the C-terminal proline residue at the P-site to restart the stalled ribosome[7]. It is now known that folding of proteins under force causes shortening of several nanometers[1, 20], which would increase the tension on the rest of the polypeptide. At the same time, the increased tension fights against this folding process. TF sits on the mouth of the ribosome in close proximity with the nascent chains folding under force, but it is unclear how TF is able to interact with these peptides to shift their folding equilibrium.

Force spectroscopy allows the unfolding and folding pathways of single proteins to be monitored in real time and offers a simple advantage over other techniques for studying the interaction of chaperones with proteins: the force can be applied specifically to the substrate while the chaperones or enzymes remain unperturbed, which is not possible using ensemble measurements[21, 22]. However, to date, studies using atomic force microscopy and optical tweezers on substrates in the presence of chaperones have not been able to directly monitor refolding events[12, 16, 23, 24]. Recent advances in single-molecule magnetic tweezers have permitted tracking the folding reaction of individual protein domains with unprecedented stability and resolution[20]. Here, we use magnetic tweezers to unfold the B1 domain of bacterial protein L to mimic the collapsed nascent chains emerging from the ribosome. Protein L is a small (62 residue) single-domain globular fold that contains no proline residues[25]. The simple topology of the fold ensures a well-defined transition state and no stable intermediates or off-pathway states[20, 26]. Protein L is therefore highly representative of the single-domain globular protein folds that emerge from the prokaryotic ribosome.

## Results

**Force spectroscopy experiments in presence of trigger factor.** We carry out force-clamp magnetic tweezers experiments on engineered polyproteins containing eight repeats of protein L, flanked at the N terminus by a HaloTag and at the C terminus by an AviTag (Fig. 1a)[20, 27]. Figure 1b shows a representative force-clamp trajectory from the protein L substrate with and without 500 μM TF in solution, a concentration at which TF saturates the substrate and the effect on folding can be readily observed. A two-pulse protocol is employed, beginning with an unfolding fingerprint pulse at high force (45 pN), followed by a refolding probe pulse at the desired force. At high forces, the eight individual domains unfold rapidly, each giving a stepwise increase in the end-to-end length of the protein (15 nm at 45 pN[20]). During the probe pulse, the polyprotein construct first experiences an instantaneous elastic recoil, as predicted by the polymer physics[28]. Next, the individual domains undergo refolding with a probability determined by the force, observed as a stepwise decrease in the length. Figure 1b shows the effect of TF on the folding probability ($P_f$) at three different forces in the physiological range. At 4.3 pN, the eight protein L domains fold rapidly and completely, regardless of the presence of TF, and the folding probability is equal to 1. At 11.9 pN, the folding probability drops to zero and no domains fold despite the high

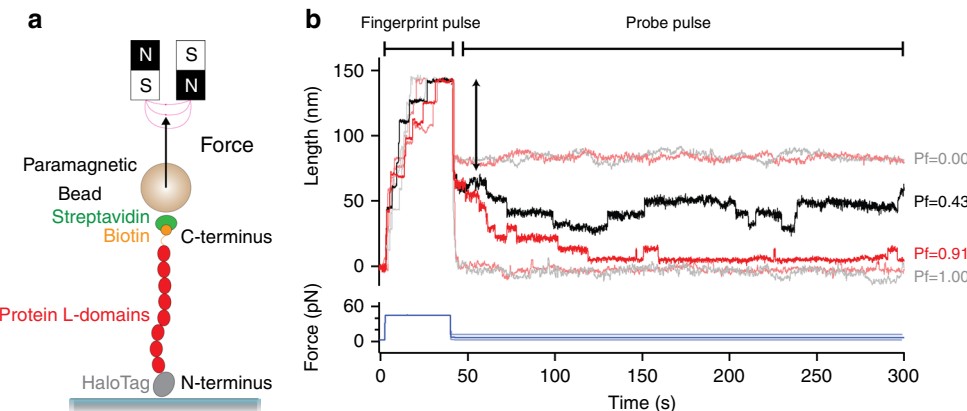

**Fig. 1** Trigger factor and the folding dynamics of protein L. **a** Schematics of the magnetic tweezers experiment, showing the octamer of protein L tethered between a glass coverslip and the paramagnetic bead. The force is applied by changing the separation of the permanent magnets and the bead. **b** Dynamics of protein L octamer at three different forces with (*red*) and without (*black*) TF. First, the protein is fully unfolded by a fingerprint pulse, where the eight unfolding events are identified as eight length steps. Second, a refolding pulse is set at a lower force (4.3, 7.4 and 11.9 pN, from *bottom to top*). At 4.3 pN (*faint color*, lowest length) all domains are able to fold by themselves, leading to a maximum probability of folding. Therefore, TF does not do any significant effect. At 11.9 pN (*faint color*, highest length), the protein is not able to refold (0 probability of folding) and TF does not either affect the probability of folding. In the intermediate force range (7.4 pN, *solid colors*) TF greatly increases the probability of folding, reflected in a higher number of folded domains

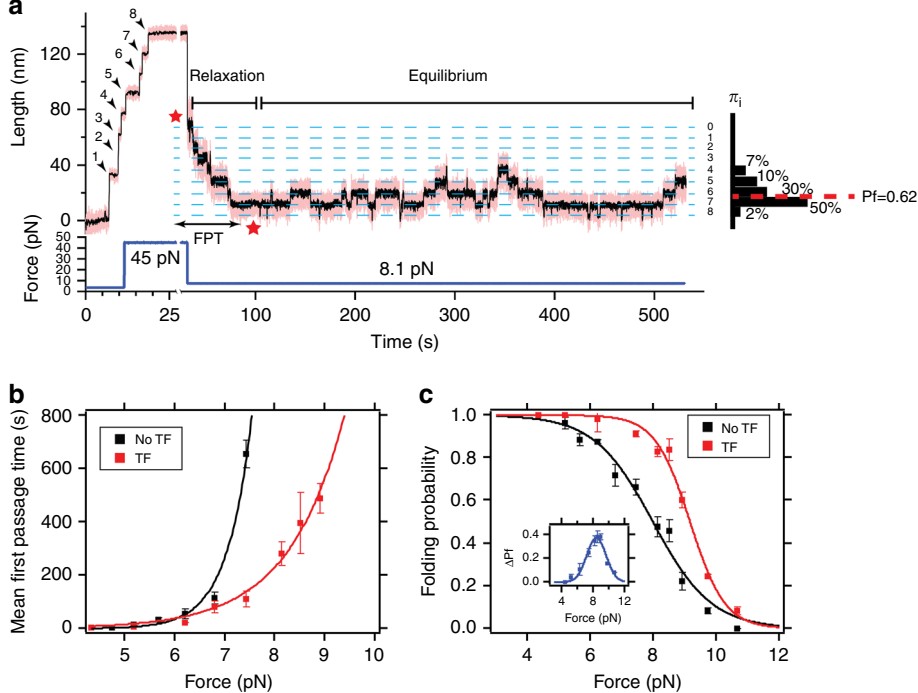

**Fig. 2** Effect of trigger factor on the folding properties of protein L under force. **a** Representative force-clamp magnetic tweezers trajectory of a protein L octamer in presence of 500 µM TF with the monitored properties highlighted. After the unfolding pulse, the molecule is set at a constant force (8.1 pN in this case), relaxing to an equilibrium state. The kinetic properties are reflected in the first passage time (FPT), time taken to, starting with all domains unfolded, fold the eight domains for the first time (time between *arrows*, *highlighted*). In equilibrium, the molecule experiences several folding-unfolding transitions, reflected in different residence times at each folding state (labeled as the number of folded domains), marked with the *dashed blue lines*. The folding probability is calculated by monitoring the fraction of time spent on each of the states (*black bars*). **b** Mean-FPT for total refolding in presence (*red*) and absence (*black*) of TF. TF modulates the folding kinetics by speeding up the time needed to refold all eight domains. Above 7.4 pN, a protein L octamer needs over 1000 s to fold completely, while TF accelerates folding over an order of magnitude. Each data point represents the average of more than ten experiments. *Error bars* represent s.e.m. **c** Folding probability in presence (*red*) and absence (*black*) of TF. The presence of the molecular chaperone TF increases considerably the folding probability over the range from 7 to 10 pN. The *inset* shows the relative increment, reaching up to 40%. Data points are calculated as described above, using >3000 s, and over more than three molecules per force. *Errors bars* are s.e.m

concentration of TF; only the elastic recoil caused by the change in force is observed. However, at an intermediate force of 7.4 pN the folding probability of the protein L construct is dramatically increased due to the presence of TF, from $P_f = 0.43$ to $P_f = 0.91$. At this force, fewer than four domains fold on average without the chaperone, but nearly all eight fold in the presence of 500 µM TF.

**Effect of trigger factor on protein folding under force.** We explore systematically the effect of TF by comparing the equilibrium and kinetic properties of protein L in the low force regime, in absence and presence of 500 µM TF. The equilibrium properties were characterized by the folding probability, as described above, and the kinetics (non-equilibrium relaxation) through the mean-first passage time (MFPT) to the totally folded state of the protein L octamer[29]. Figure 2a illustrates a representative experiment used to monitor the folding properties. After the fingerprint pulse, the protein was allowed to relax at a force between 4–12 pN, experiencing first a relaxation stage, until it reaches the equilibrium state. In equilibrium, domains fold and unfold as stepwise hopping such that detailed balance holds (every unfolding transition is counterbalanced by a refolding transition and vice versa), ensuring that the molecule is in equilibrium. For each trajectory, the FPT is calculated as the shortest time leading to complete refolding, starting from the fully unfolded state (time between the *arrows* in Fig. 2a). The MFPT is

obtained by averaging this quantity over several trajectories, providing a model-free metric that characterizes the refolding kinetics of protein L under force. Next, the folding probability is calculated from dwell-time analysis. In equilibrium, every folding state $i$ (where $i$ is the number of folded domains, shown as the *dashed blue lines* in Fig. 2a) is populated in a force-dependent way $\pi_i = t_i/t_t$, where $t_i$ is the time spent in state $i$, and $t_t = \Sigma_i t_i$ the total measuring time in equilibrium (*black bars* at the right of the trace, Fig. 2a). Thus, the folding probability is the normalized average state $P_f = \Sigma_i i\pi_i/N$, where $N = 8$ is the total number of domains ($P_f$, *blue dashed line*). Further detailed explanation on the calculation of $P_f$ can be found in Supplementary Information, with Supplementary Tables 4–10 showing explicitly $\pi_i$ for each force, in presence and absence of TF.

Figure 2b compares the MFPT in the absence (*black*) and presence of 500 µM TF (*red*), and the observed MFPTs are fit with a single-exponential function. In the presence of TF, the kinetics of folding are greatly accelerated, especially at forces above 6 pN. For example, at 7.4 pN, the protein L polyprotein needs around 600 s to reach a state where the eight domains are folded, six times slower than the time required in presence of TF (<100 s). At 8 pN, we can predict a MFPT of the order of 2500 s, while TF allows the polyprotein to refold in <300 s.

Figure 2c shows the equilibrium properties of protein L in absence (*black*) and presence of 500 µM TF (*red*) as calculated in the range from 4 to 12 pN. At any given force, the folding probability of the protein L construct is increased by the presence

of TF, and the half-point force (where $P_f$ is equal to 0.5) increases from 7.9 to 9.1 pN. In the *inset* we plot the difference between both folding probability curves in absence and presence of TF, showing that TF can increase the folding probability up to 40% at 8 pN. At the lowest forces (4–5 pN), the folding probability is close to 100%, thus the TF effect is not appreciable. It is in the intermediate force regime (5–9 pN) where TF plays the most relevant role, increasing the folding probability up to 40%. However, as the force increases, the effect of TF drops, and above 12 pN the chaperone is unable to promote folding. Non-equilibrium unfolding experiments were also performed to determine if TF affected the stability of the natively folded protein L. Because no change in unfolding rate (Supplementary Fig. 1B) or unfolding force (Supplementary Fig. 1D) was observed, we conclude that TF is released from the substrate at the time of folding and has little affinity for the folded protein L.

These results highlight the ability of TF to reshape the folding free energy landscape of protein L through interactions with the collapsed polypeptide chain. The increase in the folding probability represents an increase of the relative equilibrium free energy difference between the folded and unfolded states. TF also accelerates the refolding kinetics and keeps the unfolding kinetics unchanged. These three results combined imply that TF is indeed decreasing the stability of the unfolded state, likely by reducing its conformational entropy, a mechanism observed for other molecular chaperones[13, 30].

**Effect of trigger factor on folding at near-zero force.** This foldase activity on protein substrates under tension is independent of the known ability of TF to rescue the protein from kinetic traps (proline isomerization) or misfolded states, which has already been investigated with either bulk assays or non-equilibrium force rips[16, 17]. To further explore the activity of TF on protein L in the near-zero force regime, we design the protocol shown in Supplementary Fig. 2A. After the fingerprint pulse, where the eight domains unfold, we quench to a near-zero force (~0.7 pN) for different times $t_{quench}$, unfolding subsequently to see how many domains folded during this time. We report the fraction of folded domains as a function of the quench time, in absence and presence of TF (Supplementary Fig. 2B). Interestingly, TF does not maintain the foldase activity in this near-zero force regime. Indeed, the chaperone seems to hinder the refolding transition since, after long quench times of 20 or 30 s, all protein L domains fold in absence of TF, while the presence of the chaperone decreases the fraction to ~60%. This result is compatible with previous single-molecule studies where TF seemed to stabilize the unfolded state or to protect partially folded states[16, 18].

This intriguing ability of the mechanical force to modulate the activity of TF can be described as a consequence of the competition between the timescales of protein L folding and TF binding-unbinding. TF binds to protein substrates within times of ~$10^{-8}$ s[31], and the unbinding timescale is between 10–35 s, depending on the substrate[32]. This is much slower than folding of protein L in the absence of force, which is a good folder, folding within times of ~$10^{-2}$–$10^{-3}$ s[33]. Therefore, it is plausible to think that binding of TF to an unfolded protein L, would indeed slow down the folding dynamics, as TF has to be expelled before protein folding. Nevertheless, the presence of the pulling force greatly slows down the folding dynamics, which is in a 10–100 s timescale in the 6–10 pN range (Fig. 2b), comparable or even slower than that of TF unbinding. Therefore, upon TF binding, if this low entropy bound state is conductive to folding, and the free energy of folding is greater than the free energy of binding of the unstructured polypeptide to TF, the protein L domain will fold

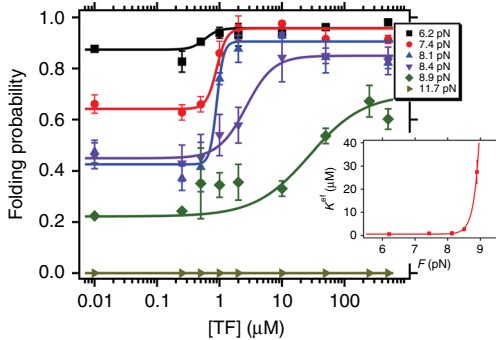

**Fig. 3** Titration curve describing the force-dependent effect of the trigger factor concentration on the folding probability of protein L. We monitor the change in the folding probability of protein L at increasing TF concentration for different forces. The sigmoidal dependence of the folding probability with the TF concentration reveals the cooperative effect of TF on protein L under force. *Inset* shows the half maximal concentration $K^{eff}_{1/2}$, showing a strong non-linear dependence of the TF effective affinity with the force. Data points are calculated as described above, using >3000 s, and over more than three molecules per force. *Error* bars are s.e.m

and expel TF within a faster timescale than that of protein L folding. This could be a possible reason that TF can work as foldase without ATP, unlike other chaperones such as HSP60 or HSP70[11, 34].

**Concentration dependency of trigger factor under force.** Cellular levels of TF are often around 50 μM, and the monomer–dimer equilibrium of unlabeled native TF has been reported to occur at 18 μM[31, 35]. Therefore, it is interesting to study the effect of the concentration of TF on the folding properties of protein L, in particular since the increase on the folding probability reported in Fig. 2 was done at a high concentration of 500 μM, well above the dimerization concentration. To investigate this, we conduct a concentration-dependency study in the force range from 6 to 12 pN (Fig. 3), where the effect of TF on folding is most prominent. At each force, the concentration of TF was increased from 0 μM (data equivalent to *black symbols* in Fig. 2c) to 500 μM (data equivalent to *red symbols* in Fig. 2c). The resulting change in the folding probability with increasing TF concentration was fit with the Hill equation (Supplementary Information for explicit equation and fitting parameters, Supplementary Table 3), which demonstrates that mechanical force modulates the effective affinity of TF for its substrate. The half maximal concentration of TF ($K_{eff}$) increases with force in a strongly non-linear way. When protein L is held at forces above 12 pN, the predicted $K_{eff}$ is so high that either TF can no longer recognize the polypeptide as a substrate, or TF cannot sufficiently reshape the free energy landscape to promote folding. These two scenarios appear indistinguishable to our experimental assay, as TF binding does not seem to induce any change along our reaction coordinate. The dependence of the step-size (length increment upon each domain unfolding) with the force follows a freely jointed chain with same contour and Kuhn length, in absence and presence of TF (Supplementary Fig. 3). However, it is possible that, as force is stretching the unfolded protein and therefore effectively changing the substrate, TF binding is very sensitive to force and at high forces it does not longer recognizes protein L as a substrate. This strong dependency of binding to protein substrates under tension has already been reported for vinculin binding to talin[36]. Indeed, the fit of the Hill sigmoid to the folding probability suggests a cooperative effect, as the change

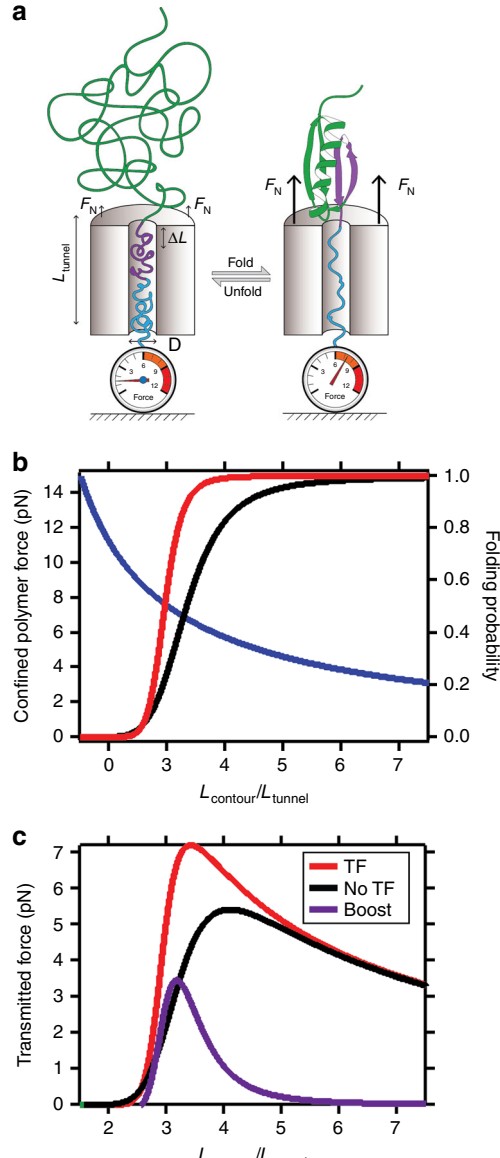

**Fig. 4** Model for trigger factor-assisted force transmission through a molecular pore. **a** A polypeptide chain exiting a molecular tunnel is under some tension due to the confined geometry (*left*). When the protein folds in the edge of the tunnel (*right*), it pulls out a fraction of the polypeptide in the channel (*purple fragment*), increasing the mechanical tension and hampering the folding transition. **b** When a polymer is confined in a tube of fixed length $L_{tunnel}$ is subject to an effective force, which depends on its contour length $L_{contour}$. This force can be approximately modeled by a freely jointed chain (FJC) model, where the end-to-end distance is fixed to $L_{tunnel}$ and the force depends on the length of the confined polymer $L_{contour}$ (*blue line*). Accordingly, the folding probability with (*red*) and without (*black*) TF of a polypeptide chain in such scenario depends on $L_{contour}$. **c** Protein folding generates shortening which increases the tension of the polymer, transmitting a force through the tunnel. We can estimate this expected force as the product between the force generated by the confined polymer and the probability of folding at such length $L_{contour}$. This is calculated with (*red*) and without (*black*) TF revealing that there is an optimal length of the confined polymer, between 3 and 4 times the length of the tunnel. Shorter polymers would induce large tensions that would avoid protein folding, so the expected force is zero. Longer polymers would lead to small values of the expected force. In the optimal region, TF boosts the transmitted force by almost 4 pN (*purple*)

takes place in a very narrow concentration range. Interestingly, this range gets broader as the force increases, revealing that this co-operativity decreases with the pulling force on the collapsed state. Below 8 pN, the change is notably steep, occurring in a range of about 1 μM, but at 8.9 pN the concentration must be increased over two orders of magnitude to reach the plateau in the folding probability. Interestingly, for most forces, a concentration lower than the dimerization concentration is enough to saturate the increase on the folding probability, which implies that monomeric TF is able to work itself as a mechanical foldase (Supplementary Fig. 4). In addition, to confirm that the variation in the folding probability is indeed due to the activity of TF and not due to any crowding effects, separate control experiments substituted TF with bovine serum albumin (BSA) and no effect was observed (Supplementary Fig. 5).

## Discussion

The increased folding probability under force due to TF assistance must relate with its biological function, in particular when it interacts with nascent chains in the mouth of the ribosome. Force is being recognized as a crucial feedback mechanism in this scenario, which is also common to other situations where proteins are translocated through molecular channels and fold on the edge of the tunnel, such as in the translocon pore[12]. However, it is still unclear how this force is transmitted through the tunnel to assist translocation, but protein folding is recognized as a source of force generation in which chaperones might participate by shifting folding to higher force values. We model this situation in Fig. 4a, showing a generic polypeptide chain translocating through a molecular pore of length $L_{tunnel}$ and diameter $D$. Due to the imposed geometric constraints, the tunnel accommodates a polypeptide chain which resembles a stretched polymer (*blue* and *purple segment* in Fig. 4a) and therefore is under an effective force[37]. Protein folding on the mouth of the channel induces a shortening of several nanometers by pulling out a fraction $\Delta$ (*purple segment*) of the confined polymer. This mechanism would transmit a force output through the channel by decreasing the length of the confined polymer to $L_{contour}$ and therefore induces an increased tension that hinders folding. TF sitting on the edge of tunnel would play a relevant role by assisting protein folding at higher forces and consequently, maximizing force transmission through the tunnel.

We can model this mechanism by combining simple polymer physics and our measurements of the folding probability, predicting what effect chaperones, such as TF, would have by interacting with the nascent chain as it folds on the edge of the ribosome[28]. We estimate the force exerted by the confined polymer with a freely jointed chain (FJC) model of Kuhn length $l_K = 0.5$ nm (measured from the elastic recoil of the unfolded polyprotein to different forces, Supplementary Fig. 6), clamped to a constant end-to-end distance and with a varying contour length $L_{contour}$ (*blue line*, Fig. 4b)[38]. Therefore, shorter polypeptides transmit higher forces through the tunnel. In turn, the escaped unfolded protein would have a probability of folding that depends on the length of the confined polymer $L_{contour}$ (*blue segment* in Fig. 4a). This is plotted in presence and absence of TF (*red* and *black* respectively, Fig. 4b). If $L_{contour}$ is such that folding is allowed, the force feedback would be transmitted through the tunnel whereas, if this force is too high, the system would likely incorporate the shortening segment $\Delta$ back inside the tunnel. On average, the system transmits a force due to protein folding that can be estimated as the product between the folding probability and the force exerted by the confined polymer. We plot the expected force in presence (*red*) and absence (*black*) of TF (Fig. 4c). Our simple model of force transmission through

molecular channels leads to two relevant implications. First, the presence of the chaperone increases the expected force output due to protein folding by up to 4 pN (*purple*). As discussed previously, this value is of relevance in many biological situations such as the rescuing of stalled polypeptide chains in the translation process. Indeed, the force needed for this process has been estimated in the 7–9 pN range[7]. Second, our studies imply that there is an optimal length range of the confined polypeptide that meets the compromise between transmitting an optimal force value and allowing the protein to fold in the mouth of the tunnel. We discover that the optimal polymer contour length must be ~3 times the length of the channel, about 25 nm, meaning that when ~60 residues are accommodated inside the ribosomal tunnel, maximum force output is generated. Notably, the ribosomal exit tunnel can accommodate 30 amino acids inside the tunnel when in a random coil configuration, or up to 60 amino acids if they begin to take on alpha-helical structures[39]. These values are in good agreement with our estimations, in particular considering the coarse approximations we have assumed. The actual force-to-extension relation depends on several unaccounted factors such as the particular geometry of the ribosomal exit tunnel[40], the shape of the polypeptide chain (which might reduce the effective diameter and lead to higher confinement)[41], or possible interactions within the polymer or with the walls. These factors can reduce the effective stiffness of the polypeptide, shifting the force curve in Fig. 4c to shorter polypeptide lengths.

Overall, we have measured the folding probability of a protein substrate under force in presence of a molecular chaperone. The increased folding probability and accelerated folding kinetics suggest that TF reshapes the folding landscape likely by reducing the entropy of the unfolded state. In addition, the action of TF is strongly force dependent, as its effective affinity decreases with the pulling force so that above 12 pN no effect can be observed. This scenario is of particular importance for force transmission through molecular channels. The shortening due to protein folding can be the mechanism by which this force is transmitted; therefore, the presence of chaperones such as TF would maximize this value by allowing folding at higher forces. Taken together, these findings indicate that TF has a mechanical foldase activity for the small, globular fold of protein L, which adds yet another dimension to the ability of TF to assist protein folding.

## Methods

**Magnetic tweezers experiments**. The protein constructs were engineered as polyproteins with eight repeats of the protein L B1 domain, flanked at the N-terminal and C-terminal with the HaloTag and AviTag, respectively. TF was expressed and purified by the Kalodimos Lab (University of Minnesota). BSA was obtained from Sigma-Aldrich. Paramagnetic Dynabeads (M-270) coated with streptavidin having a diameter of 2.8 μm were purchased from Invitrogen.

Magnetic tweezers experiments are carried out on an inverted microscope (Olympus IX-71/Zeiss Axiovert S100) mounted on a nanofocusing piezo actuator (P-725; Physik Instrumente), a magnet-positioning voice coil (LFA-2010, Equipment Solutions), and a high-speed camera (AVT Pike F-032B, Allied Vision Technologies). The position of the magnets was controlled with a linear voice-coil with a speed of ~0.7 m/s speed and 150 nm position resolution.

Magnetic tweezers experiments are performed in fluid chambers made of two sandwiched cover slides separated by two strips of parafilm. The bottom and top cover slides were cleaned by sonication for 60 min in 1.5% Hellmanex solution, 30 min in acetone, and 30 min in ethanol. After sonication the bottom slides are silanized by immersion for 40 min in a solution of (3-aminopropyl)-trimethoxysilane 0.1% v/v, in ethanol and then dried at 100 °C for >1 h. After sandwiching both the cover slips the chamber is functionalized with glutaraldehyde, nonmagnetic polystyrene beads (2.8 μm) and finally with HaloTag (O4) amine ligand. All magnetic tweezers experiments were carried out at 22 °C in phosphate buffered saline pH 7.4. Data acquisition and analysis was carried out in Igor Pro (Wavemetrics). All data employed in the text was obtained from traces showing full-fingerprint (eight unfolding steps during the fingerprint pulse, Fig. 2a).

**Calculation of the folding probability**. We calculate the folding probability from the equilibrium stage at a particular force. This stage is different from the relaxation stage (non-equilibrium) as detailed balance condition holds. We label each state by $i$, the number of folded domains. Thus, the length trajectory can be discretized as the time series of each state $i(t)$, where $i = 0, 1,…, 8$. The folding probability is calculated from the populations of each state at a particular force $\pi_i$. Given a force $F$, we calculate the population of each state as the fraction of time the time has spent on it, so $\pi_i = t_i/t_t$, where $t_i$ is the time spent on state $i$ and $t_t$ the total observation time at force $F$. For each force, we combine several trajectories coming from different molecules, in order to improve the statistical significance of our measurements. Therefore, $t_t$ is a combination of several sampling pulses and molecules. Then, the folding probability $P_f$ is calculated as:

$$P_f = \sum_{i=0}^{8} i\pi_i. \tag{1}$$

**Unfolding experiments of protein L**. Protein L molecules are unfolded in two different protocols, by applying a constant force (force-clamp) or by applying a linearly increasing force (force-ramp). In force-clamp experiments the force is kept constant at 15, 26, and 35 pN and the unfolding kinetics by MFPT measurements to the fully unfolded octamer. For the force-ramp experiments the force is increased linearly, we start at a force of 4 pN, and pull at 1.5 pN/s until reaching 64 pN.

**Sigmoid fits to the folding probability**. The folding probability as a function of the force is fitted with a sigmoid equation:

$$P_f(F) = B + \frac{M}{1 + e^{-\frac{(F - F_m)}{r}}}, \tag{2}$$

where $B$ and $M$ are, respectively, the base and maximum value of the sigmoid (here fixed to $B = 1$ and $M = -1$), $F_m$ is the force at which the half change is achieved and $r$ is the rate.

**Exponential fits to MFPT**. The MFPT as a function of the force is fit with an exponential growth function:

$$\text{MFPT}(F) = A e^{(-F/r)}. \tag{3}$$

**Hill fits to the concentration-dependency study**. The change of folding probability with the TF concentration is fit with a Hill equation:

$$P_f([\text{TF}]) = B + \frac{M - B}{1 + ([\text{TF}]_m/[\text{TF}])^r}, \tag{4}$$

where $B$ and $M$ are, respectively, the base and maximum $P_f$, $[\text{TF}]_m$ is the concentration of the half change and $r$ the rate.

**Kuhn length measurement of the unfolded protein-L octamer**. We measure the Kuhn length of the unfolded protein-L octamer by measuring the elastic recoil starting from a reference force of $F_R = 45$ pN to different probe forces $F$. Assuming that the unfolded octamer behaves as a freely jointed chain, the elastic recoil $\Delta L(F_R, F)$ follows:

$$\Delta L(F_R, F) = L_c \left[ \left( \coth \frac{F_R l_K}{kT} - \frac{kT}{F_R l_K} \right) - \left( \coth \frac{F l_K}{kT} - \frac{kT}{F l_K} \right) \right], \tag{5}$$

where $L_c$ is the contour length of the molecule, $l_K$ its Kuhn length and $kT = 4.11$ pN nm the thermal energy. For each individual molecule, we use Eq. (5) to estimate the contour length $L_c$ of each molecule. Then we can normalize the elastic recoil of each molecule and calculate the relative recoil $\Delta L(F_R, F)/L_c$.

**Step-sizes in presence and absence of TF**. Upon protein unfolding, each domain, from the native state, extends and relaxes to an equilibrium length determined by the pulling force. We refer to this increment in length as the step-size and it can be well represented by a freely jointed chain model, so:

$$\Delta L(F) = L_c \left[ \coth \left( \frac{F l_K}{kT} \right) - \frac{kT}{F l_K} \right] \tag{6}$$

where $kT = 4.11$ pN nm at room temperature, $L_c$ is the contour length of the polymer (length at infinite force) and $l_K$ the Kuhn length (length of the "Kuhn segments") which relates to the stiffness of the polymer.

**Data availability**. All relevant data are available from the corresponding authors upon reasonable request.

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

## Acknowledgements

This work was supported by NSF Grant DBI-1252857, and by NIH Grants GM116122 and HL061228. RTR acknowledges Fundacion Ramon Areces for financial support. E.C.E. was supported by NIH F30-HL129662. We would like to acknowledge Professor Charalampos Kalodimos and Dr Chengdong Huang from University of Minnesota for sharing TF with us. We would like to thank all the members of Fernandez lab for their valuable comments on the manuscript.

## Author contributions

S.H., R.T.-R., E.C.E. and J.M.F. designed the project, S.H., R.T.-R., and J.V.-O. performed the experiments and analyzed the data, S.H., R.T.-R., E.C.E., and J.M.F. wrote the paper.

## Additional information

**Competing interests:** The authors declare no competing financial interests.

