## [Peer Review file · Nature Communications]

Reviewers' comments:

Reviewer #1 (Remarks to the Author):

This manuscript by Hadar et al. reports on the effects of mechanical tension on the activity of E. coli trigger factor (TF). Serially-connected protein L domains, total eight, were unfolded by high force at 45 pN and refolding of these domains was induced by relaxing the tension to a low pN value. TF had an effect of increasing the probability of observing the folded state. The authors thus claimed that TF works as a foldase that shifts the balance of the folding energy landscape to favor the folded state. By modeling the force exerted by the confined polymer, the authors then claimed that TF on average transmits higher force to the nascent peptide chain than in the absence of TF by assisting folding, especially in the mid-force range. The experiments are well designed, and the presented data sets would be of interest to both fields of single-molecule force spectroscopy and protein chaperones. However, as stated in the major comments, the potential of the study has not been fully explored. Therefore, this reviewer finds that the manuscript needs to be much improved before final recommendation for publication in Nature Communications.

Major comments

1. How did the author calculate the folding probability in Fig. 1B and Fig. 2C? The authors need to specify the detailed method. Since the authors monitored unfolding and refolding kinetics of protein L domains in an equilibrium state (Figure 2), it should be possible to obtain some quantitative values related to the folding energy landscape, rather than simply calculating the folding probability. It is needed to show that the free energy difference between the folded and unfolded states (ΔG) at different force levels, and that how these ΔG values are changed in the presence of TF. Such data set would more directly support the conclusion that the TF is a mechanical foldase.

2. The folding probability data in Fig. 2C shows that TF cannot induce folding of the protein L domains when the mechanical tension is above 12 pN. There can be two scenarios (also briefly stated in page 7 of the manuscript). TF may not be able to bind any more to these highly-stretched unfolded substrates, or TF can bind to the substrate but cannot accelerate its folding. Can the authors distinguish between these possibilities?

3. The TF titration data in Figure 3 is interesting. In particular, as the authors noted, the data below 8.1 pN showed sigmoidal enhancement of folding around 1 μ M TF, which was most evident at 8.1 pN. However, only 0.3 pN increase in the tension significantly dampened the sigmoidal enhancement. The folding probability curve of the protein L domains in the absence of TF (shown in Fig. 1C, black curve) does not show any discontinuities in the corresponding region. Thus, it seems likely that the cooperative effects between multiple TFs, not changes in the protein L substrates, are responsible for the sigmoidal behavior. Since the role of dimeric TFs has been controversial, the authors could attempt to investigate the catalytic activity of dimers vs monomers by disrupting the dimer interface and/or crosslinking TFs to induce dimers and investigating its activity at low concentrations. In this sense, this comment is also related to the Comment 2, which asked whether the binding or activity of TFs is affected by the mechanical tension.

4. In Figure 4, the authors used a Kuhn length of 0.47 nm. For a long chain, where the contour length is much longer than the Kuhn length, a half of the Kuhn length corresponds to the persistence length of the worm-like chain model. This gives a persistence length of 0.235 nm, shorter than the value of 0.4 nm typically used in the field. The authors need to show that the conclusions remain valid after increasing the Kuhn length up to 0.4 nm. Increasing the persistence length may significantly decrease the internal tension in the nascent chain because it largely arises from the entropic cost.

Minor comments

1. How did the authors distinguish the relaxation phase from the equilibrium in Figure 2A?
2. In Page 5, the FPT seems to indicate the time between the arrows, not the stars.
3. While the authors provided an experiment with BSA ruling out the crowding effect, a more convincing test of the activity would be disrupting the interaction between TF and peptide either by introducing a mutation in TF, or by using FK506 that is known to interfere with peptide binding of TF. (ref: Patzelt et al., "Binding specificity of Escherichia coli trigger factor", PNAS, 2001.)

Reviewer #2 (Remarks to the Author):

The authors studied refolding of a tandem of 8 L-proteins using magnetic tweezers in the presence and absence of TF. The observed increase in folding probability is then explained by a model that assumes binding of TF to unfolded chain. The study was focused mostly on 500 μ M TF concentration and the tethers were made via commonly used strong linkages (HaloTag and Streptavidin-biotin), allowing them to do analysis for long time scales.

While motivating the study, the author claim that "However, to date, studies using atomic force microscopy and optical tweezers on substrates in the presence of chaperones have not been able to directly monitor refolding events". The phrase suggests that the current paper provides technical advancements related to single molecule pulling assays, which is not true. Single molecule investigation of chaperone assisted folding is an emerging field and lack of such studies in the literature is not due to technical limitations. One recent example, is a new report that studied HSP70 action on a protein substrate under force (see PMID 27783598).

For the reported data to be relevant, the timescales reported in this study need to be compared with the timescale of TF binding to translating and non-translating ribosome and nascent chains.

The protein used in this study is 62 residues long. Considering the polypeptide length that the ribosome tunnel can accommodate and the size of the TF when bound to ribosome, I do not really believe that the process described in this study is relevant to protein L folding in cells.

We know that substrate-bound TF could leave the ribosome while preserving its expanded state (see e.g. PMID: 17051157). How does the current study relates to this phenomenon ?

The authors argue that binding of hydrophobic core of protein L to a single hydrophobic patch of TF, reduces the entropy of the chain, thereby promotes folding. The authors then claim that this could explain why TF can be a foldase without using ATP while HSP70 and HSP90 need ATP to catalyze folding. And finally, the authors presented the results as evidences for the ability of TF to reshape the folding free energy landscape of protein L. I have a few comments: (1) if TF acts as a foldase under force because of the entropic cost, we would expect this to be even more of a case in the absence of force. In the latter, the entropy cost would be larger and folding would be favoured subsequently. (2) The claim that HSP70 cannot bind unfolded chains without ATP cycle is incorrect. (3) There are many holdases among cellular chaperones and they are typically known to suppress folding. This might seem inconsistent with the model proposed here. (4) The author might want to consider interaction between TF and loosely folded protein L as an alternative model that could explain the results. The fact that the unfolding forces of the fully folded protein L did not rise in the presence of TF does not necessarily rule out a model that TF promotes folding by transiently protecting a partially folded state. (5) It would be interesting to see how the authors put their findings in the context of protein translocation literature. (6) Given the proposed model, I am wondering why binding of TF

to a chain would not introduce correlation into folding of neighbouring proteins. TF has a patchy interface that binds both hydrophobic and hydrophilic chains. (7) I am surprised that the authors highlighted the changes to the folding landscape. I would not really call this "reshaping". Nothing has changed "during" folding. And by the way, TF has been long known to bind chains, from both single molecule and bulk studies. (8) In the entropic picture proposed above, the entropic terms associated with TF dynamics as well as the enthalpic terms associated with dimerization of TF have not been taken into consideration. See PMID PMID: 23565160.

One concern I have is that the major part of this study is based on 500 uM concentration of TF, i.e. the dimeric regime, while the whole study has been interpreted in the context of ribosome associated TF and translocation. Ribosome bound to TF is very different in terms of its conformation and available interacting surface with soluble monomeric TF and is obviously not comparable with dimeric TF. At concentrations of a few hundred nM the authors did not see significant effects. How do the authors respond to this?

I found the theoretical study interesting for polymer physicist but I believe it is too simplistic for the purpose of this paper. The biological complexity of the system has not been taken into account. Additionally, the force applied in the experiment to the two ends of the protein L differs from the configuration used in the model.

Minor

Page 8, Line 2: Please correct for spelling error in "though".

Reviewer #3 (Remarks to the Author):

The present study aims at elucidating the mechanism of action of the molecular chaperon TF (Trigger Factor). The authors use magnetic tweezers to monitor folding of protein L under mechanical force in the presence of different concentrations of TF. The results of these studies suggest that TF facilitates the folding of protein L in the physiological force range of 4-10 pN by reducing the conformational entropy of the unfolded state. Moreover, the results show that the foldase activity of TF is force dependent as the effective affinity of TF for protein L decreases at higher forces. I found this work original, and the reported data elegant and carefully analyzed, supporting the major conclusions drawn from them. The manuscript is generally well written and clear. Consequently, I recommend this work for publication in Nature Communications, after revision to address the following minor points

1. On page 8 the authors write "Notably, the ribosomal exit tunnel can accommodate 35 amino acids inside the tunnel[33]". In reality, reference 33 suggests 30 amino acids in the ribosomal tunnel. Thus "35 amino acids" should be changed into "30 amino acids".
2. At the end of page 8, the authors make a list of "several uncounted factors" that can affect the "actual force-to-extension relation". I suggest quoting at least one reference for each suggested factor to support their claims.

Reviewer #1 (Remarks to the Author):

This manuscript by Hadar et al. reports on the effects of mechanical tension on the activity of E. coli trigger factor (TF). Serially-connected protein L domains, total eight, were unfolded by high force at 45 pN and refolding of these domains was induced by relaxing the tension to a low pN value. TF had an effect of increasing the probability of observing the folded state. The authors thus claimed that TF works as a foldase that shifts the balance of the folding energy landscape to favor the folded state. By modeling the force exerted by the confined polymer, the authors then claimed that TF on average transmits higher force to the nascent peptide chain than in the absence of TF by assisting folding, especially in the mid-force range. The experiments are well designed, and the presented data sets would be of interest to both fields of single-molecule force spectroscopy and protein chaperones. However, as stated in the major comments, the potential of the study has not been fully explored. Therefore, this reviewer finds that the manuscript needs to be much improved before final recommendation for publication in Nature Communications.

Major comments

1. How did the author calculate the folding probability in Fig. 1B and Fig. 2C? The authors need to specify the detailed method. Since the authors monitored unfolding and refolding kinetics of protein L domains in an equilibrium state (Figure 2), it should be possible to obtain some quantitative values related to the folding energy landscape, rather than simply calculating the folding probability. It is needed to show that the free energy difference between the folded and unfolded states (ΔG) at different force levels, and that how these ΔG values are changed in the presence of TF. Such data set would more directly support the conclusion that the TF is a mechanical foldase.

Previous version of the manuscript includes a brief explanation on how we calculate the folding probability P_f . We agree that it might be insufficient to fully understand our methodology. We have now extended our explanation with a new detailed section in the SI. The central idea behind our method is to estimate the occupancy π_i of each folding state from the force-clamp equilibrium trajectories in order to calculate the average folding state (average number of folded domains at a particular force) and normalize it by the number of domains in the polyprotein (8 in our case). We have detailed this methodology in the SI and also included the explicit tables showing π_i at each force in presence and absence of TF (Table S4A-G).

The folding probability is a direct measure of the equilibrium free energy difference between the folded and unfolded state, if you assume each domain of the polyprotein to behave independently and with well-defined folded and unfolded states. Therefore, the free energy difference between folded-unfolded states is a magnitude that is implicit in the folding probability (actually it is just a change of variable). Nevertheless, we agree with the reviewer that the free energy provides a more direct intuition on the effect of TF on the free energy landscape of the molecule. We have now included a section in the SI plotting the free energy difference between the folded and unfolded states as a function of the pulling force. This is simply calculated from the relative population of the folded (P_f) –equivalent to the folding probability- and unfolded ($P_u=1-P_f$) states. Figure S2 shows that the free energy difference between the folded-unfolded state increases in presence of TF (in absolute value), meaning that TF decreases the stability of the unfolded state (as the folding kinetics are accelerated and the unfolding kinetics unchanged).

The mechanical foldase activity from the free energy difference point of view has been further discussed in the main text:

“These results highlight the ability of TF to reshape the folding free energy landscape of protein L through interactions with the collapsed polypeptide chain. The folding probability is an equilibrium measure of the relative free energy difference between the folded and unfolded states. Indeed, assuming that each domain folds independently and with

well-defined folded and unfolded states, it can be calculated as $\Delta G_{U-F}(F) = -kT \log P_f / (1 - P_f)$. To highlight the ability of TF to increase this magnitude, Fig. S2 shows the free energy difference between the folded and unfolded states as a function of the force, in presence and absence of TF. Besides increasing the folding probability, TF accelerates the refolding kinetics, keeping the unfolding kinetics unchanged. These three results combined imply that TF is indeed decreasing the stability of the unfolded state, likely by reducing its conformational entropy, a mechanism observed for other molecular chaperones [13, 30]."

2. The folding probability data in Fig. 2C shows that TF cannot induce folding of the protein L domains when the mechanical tension is above 12 pN. There can be two scenarios (also briefly stated in page 7 of the manuscript). TF may not be able to bind any more to these highly-stretched unfolded substrates, or TF can bind to the substrate but cannot accelerate its folding. Can the authors distinguish between these possibilities?

In our force-clamp protocol, we monitor the relative length of a single molecule as function of time for a particular force value. Therefore, we are not able to observe *directly* the binding of TF to its substrate, but to observe *indirectly* its effect on the folding properties of the protein, unless it induces some conformational change along this reaction coordinate. Nevertheless, we do not observe any change in the unfolded molecule properties when TF is present. In order to provide a quantitative evidence of this, we have included a new section in the SI with new experimental data, where we show the step size and the relative contraction of the unfolded molecule (Figures S4 and S7). Clearly, the polymer properties of the unfolded molecule remain unchanged when TF is present. Therefore, the two mentioned scenarios cannot be distinguished.

An interesting aspect of our experimental system is that the pulling force is actually changing the substrate by stretching it in a non-linear fashion, given the force-extension relation imposed by the polymer properties of the unfolded molecule. Indeed, force is known to have a very strong influence on the binding properties of molecules to tensioned substrates. For example, vinculin binds talin at forces >5 pN, but unbinds at forces >25 pN, when it does no longer recognizes its substrate due to the conformational changes induced by the force (M. Yao et.al. Scientific Reports, 2014). In our particular case, TF seems to interact just with the unfolded state, as the stability of the folded state remains unchanged, while the unfolded state is destabilized. The unfolded state is being continuously changed by the pulling force, which stretches it. Indeed, most of the change in the end-to-end distance occurs up to 12 pN, when the molecule reaches $\sim 63\%$ of its contour length (see Fig. S4). Therefore, we hypothesize that TF cannot recognize highly stretched polymers as a substrate, requiring a certain conformational freedom for the unfolded protein to accommodate on its groove, respecting the constraints imposed by the force.

A further discussion on this point has been now included in the manuscript:

"These two scenarios appear indistinguishable to our experimental assay, as TF binding does not seem to induce any change along our reaction coordinate. The dependence of the step-size (length increment upon each domain unfolding) with the force follows a Freely-Jointed chain with same contour and Kuhn length, in absence and presence of TF (see SI and Figure S4). However, it is possible that, as force is stretching the unfolded protein and therefore effectively changing the substrate, TF binding is very sensitive to force and at high forces it does not longer recognizes protein L as a substrate. This strong dependency of binding to protein substrates under tension has already been reported for vinculin binding to talin [36]."

3. The TF titration data in Figure 3 is interesting. In particular, as the authors noted, the data below 8.1 pN showed sigmoidal enhancement of folding around 1 μ M TF, which was most evident at 8.1 pN. However,

only 0.3 pN increase in the tension significantly dampened the sigmoidal enhancement. The folding probability curve of the protein L domains in the absence of TF (shown in Fig. 1C, black curve) does not show any discontinuities in the corresponding region. Thus, it seems likely that the cooperative effects between multiple TFs, not changes in the protein L substrates, are responsible for the sigmoidal behavior. Since the role of dimeric TFs has been controversial, the authors could attempt to investigate the catalytic activity of dimers vs monomers by disrupting the dimer interface and/or crosslinking TFs to induce dimers and investigating its activity at low concentrations. In this sense, this comment is also related to the Comment 2, which asked whether the binding or activity of TFs is affected by the mechanical tension.

We appreciate the comment of the reviewer. We agree that the possible role of TF dimer on the folding enhancement under force is an important aspect of our results that was not sufficiently discussed in our previous version of the manuscript. In our initial study on TF, we used a concentration of 500 μM (Figure 2), which clearly showed the mechanical foldase activity of TF. Hence, we address the titration experiments, trying to investigate whether the dimeric or the monomeric forms of TF are needed for the folding enhancement under force.

However, TF undergoes a monomer-dimer equilibrium in solution with an approximate K_d of 18 μM (Patzelt et al. *Biological Chemistry*, 2002, Saio et.al. *Science*, 2014). Furthermore, we obtained the TF from Prof. Kalodimos Lab, from University of Minnesota, who confirmed this dimerization K_d . As it can be clearly seen in Fig. 3, the main change in the folding probability occurs at concentrations below 2 μM , meaning that the monomeric form of TF has the mechanical foldase activity. Just at higher forces (above 8.5 μM), higher concentrations of TF are needed to enhance folding. In order to illustrate more clearly this point, we include Fig. S5, where we plot the folding probability versus the force at four different concentrations (0, 1, 2, and 500 μM). The biggest shift on the folding probability occurs up to 1 μM , well below the dimerization K_d . After that, significant increases on the TF concentration just increase the folding probability at high forces. Therefore, we do not believe the experiments suggested by the reviewer would provide any further clarification on the mechanical foldase activity of monomeric TF, which is clearly shown in our data.

We think that the sigmoidal and nonlinear behavior of the $K_{1/2}$ (concentration to reach half the maximal folding enhancement), is related to the strong nonlinear dependence of the folding probability with the pulling force and the possible force dependence of TF binding to substrates under force. The folding probability is already very sensitive to force changes, especially in the range from 7 – 9 pN, where in about 2 pN force change, the folding probability drops $\sim 50\%$. This is also reflected in the change populations of the folding states π_i , as see in tables S4A-G. This is related to the strong dependence of the end-to-end distance of the unfolded molecule with the pulling force, extends about $\sim 60\%$ of its contour length from 0 to 10 pN. We have included further comments on the possible role of the dimer, stating that the monomer has itself a mechanical foldase activity.

4. In Figure 4, the authors used a Kuhn length of 0.47 nm. For a long chain, where the contour length is much longer than the Kuhn length, a half of the Kuhn length corresponds to the persistence length of the worm-like chain model. This gives a persistence length of 0.235 nm, shorter than the value of 0.4 nm typically used in the field. The authors needs to show that the conclusions remain valid after increasing the Kuhn length up to 0.4 nm. Increasing the persistence length may significantly decrease the internal tension in the nascent chain because it largely arises from the entropic cost.

The Kuhn length (or the persistence length) of an unfolded molecule depends on a large number of factors, which are generally uncontrolled or difficult to access experimentally (*Statistical Mechanics of Chain Molecules*, P. J. Flory, Hanser Publishers). We are aware that, in the field of force spectroscopy, a persistence length of 0.4 nm is widely accepted. However, for polyproteins, it is unclear how does this value depends on the number of folded/unfolded domains or on its particular configuration. For the argument of our discussion, we use a fully unfolded protein. We have determined the Kuhn length of a fully unfolded protein by monitoring the elastic recoil at different force changes. Starting with the protein unfolded at a high force of 45 pN, we quench the force to different values and measure the contraction due to the polymer collapse. Fitting to a Freely-Jointed chain, we obtain a Kuhn length of 0.5 nm. We have

repeated this experiments in presence of TF, finding no appreciable change in the polymer properties of the unfolded protein, obtaining a Kuhn length of ~ 0.6 nm (Figure S7). We use this value for our calculations on Fig. 4.

Nevertheless, the conclusions which rise from the discussion on Fig. 4 are and not too sensitive to the particular value of the Kuhn length, if it is within the values reasonable for unfolded proteins. For example, using the Kuhn length of ~ 0.5 nm, we estimate that the optimal transmitted force results when the confined polymer has a length of ~ 3.4 times that of the tunnel. Increasing the Kuhn length to 1.1 nm shifts the optimal length to a higher value of ~ 3.8 times the length of the tunnel, so not an appreciable change. Therefore, the main conclusions extracted from this discussion are maintained upon reasonable Kuhn length changes. This is because the transmitted force is obtained by multiplying the folding probability and the Freely Jointed Chain equation. The steepest behavior comes from the folding probability, which dominates the shape of the transmitted force.

Minor comments

1. How did the authors distinguish the relaxation phase from the equilibrium in Figure 2A?

Upon force decrease from the unfolded state, the molecule relaxes to an equilibrium state that depends on the force. In contrast with the relaxation phase (characterized mainly by downward refolding steps), during the equilibrium phase, the detailed balance condition holds, which is generally accepted as a sufficient condition for equilibrium (*“Stochastic Processes in Physics and Chemistry”*, Van Kampen, North Holland, 2007). This means that every transition from state $i \rightarrow j$ is counterbalanced by a transition from $j \rightarrow i$. We just include the equilibrium phase to the calculation of the folding probability, while the relaxation phase includes information on the kinetics of the system, represented by the MFPT. We explicitly mention this in the manuscript:

“In equilibrium, domains fold and unfold as stepwise hopping such that detailed balance holds (every unfolding transition is counterbalanced by a refolding transition and vice versa), ensuring that the molecule is in equilibrium.”

2. In Page 5, the FPT seems to indicate the time between the arrows, not the stars.

We apologize for the typo. It has been corrected.

3. While the authors provided an experiment with BSA ruling out the crowding effect, a more convincing test of the activity would be disrupting the interaction between TF and peptide either by introducing a mutation in TF, or by using FK506 that is known to interfere with peptide binding of TF. (ref: Patzelt et al., "Binding specificity of Escherichia coli trigger factor", PNAS, 2001.)

BSA is commonly used as a sufficient control experiment in similar assays, see (Mashagui et.al. Nature, 2013; or Mannini et.al. PNAS, 2012). We agree with the reviewer that blocking the TF activity with some specific peptide would provide a strong evidence on its particular mechanism, but we believe that study would go beyond the present work, whose focus is to show for the first time the mechanical foldase activity of a molecular chaperone.

Reviewer #2 (Remarks to the Author):

The authors studied refolding of a tandem of 8 L-proteins using magnetic tweezers in the presence and absence of TF. The observed increase in folding probability is then explained by a model that assumes binding of TF to unfolded chain. The study was focused mostly on 500 μ M TF concentration and the

tethers were made via commonly used strong linkages (HaloTag and Streptavidin-biotin), allowing them to do analysis for long time scales.

While motivating the study, the author claim that “However, to date, studies using atomic force microscopy and optical tweezers on substrates in the presence of chaperones have not been able to directly monitor refolding events”. The phrase suggests that the current paper provides technical advancements related to single molecule pulling assays, which is not true. Single molecule investigation of chaperone assisted folding is an emerging field and lack of such studies in the literature is not due to technical limitations. One recent example, is a new report that studied HSP70 action on a protein substrate under force (see PMID 27783598).

We apologize for the confusion. Our intention is not to suggest that we provide any technical advance related to single molecule force spectroscopy techniques. We are aware of the work of B. Bukau and S. Tans on the study of the influence of chaperones such as TF or HSP70 using single molecule techniques (which we indeed cite in our manuscript, see reference 16 and 24), but, to our knowledge, none has monitored *directly* refolding events. By “directly”, we mean observation of the shortening of the peptide due to the protein collapse and folding transition. Notice the absence of such events in the relaxation trajectories of maltose binding protein in the work by Mashaghi et al. Nature 2013.

In contrast, the focus of our paper is to show, for first time, the influence of a molecular chaperone such as TF on the folding properties of a protein under force. Our data clearly shows that TF is able to increase the folding probability and to accelerate the folding kinetics of protein L domains under a force. Hence, we describe a new possible activity for molecular chaperones, the mechanical foldase. The strongest influence of TF is in the range of 4-10 pN, which can be referred to as the “physiological force range” meaning that forces, which appear at the molecule level inside the cell are in this range (see for example for the case of titin Rivas-Pardo Cell Reports, 2016; or for the case of the ClpXP motor: Aubi-Tam, et.al. Cell, 2011; or M. Yao et.al. Scientific Reports, 2014; for the case of Talin). This new activity can be of relevance in many cellular processes. In particular, we put it in the context of protein translocation, through molecular pores, where the confinement imposes a mechanical tension, especially in the case of nascent chains through the ribosomal tunnel, where they actually meet TF which interact with it.

For the reported data to be relevant, the timescales reported in this study need to be compared with the timescale of TF binding to translating and non-translating ribosome and nascent chains.

We agree with the reviewer that the implied timescales in the folding-unfolding of protein L and binding-unbinding of TF are very relevant for our study. We thank the reviewer for the suggestion. In our study, we measured the refolding time for complete refolding of an octamer protein L, finding it to be in the range of 1-400 seconds depending on the applied force (see Fig. 2B). So at 6.2, 6.8, 7.4 and 8.5 pN it takes 22, 81, 110 and 397 seconds respectively on average for complete refolding. Therefore, the refolding time is strongly dependent on the pulling force, but we can estimate that a single protein domain would fold in a 10 second timescale when the force is between 4-10 pN (ranging from ~ 1 s at lowest force to $\sim 10^2$ s at higher forces). Kaiser et al. show that TF is associated with the nascent chain for around 10-35 seconds (Kaiser et al. Nature, 2006), depending on the hydrophobicity of the substrate. Therefore, in the presence of a pulling force, the timescale for folding is comparable to that of TF binding.

We have now included an extended discussion on the implied biological timescales and how they are relevant for our findings:

“TF binds to protein substrates within times of $\sim 10^{-8}$ seconds [31], and the unbinding timescale is between 10 – 35 seconds, depending on the substrate [32]. This is much slower than folding of protein L in the absence of force, which is a good folder, folding within times of $\sim 10^{-2}$ - 10^{-3} seconds [33]. Therefore, it is plausible to think that binding of TF to an unfolded protein L, would indeed slow down the folding dynamics, as TF has to be expelled before protein folding. Nevertheless, the presence of the pulling force greatly slows down the folding dynamics, which is in a 10 – 100 second timescale in the 6 – 10 pN range (see Fig. 2B), comparable or even slower than that of TF unbinding.”

The protein used in this study is 62 residues long. Considering the polypeptide length that the ribosome tunnel can accommodate and the size of the TF when bound to ribosome, I do not really believe that the process described in this study is relevant to protein L folding in cells.

Protein L has been studied extensively as a model protein to understand protein folding by David Baker and others since 1996 (Scalley, et. al. *Biochemistry*, 1997; Scalley, et. al. *PNAS* 1996; Gu et al. *JMB*, 1997; Plaxco et. al. *Nature Struct. Biol.*, 1999; Plaxco et. al. *PNAS*, 1999; Maity et. al. *J. Am. Chem. Soc.* 2016; Tae Yeon et. al. *J. Mol. Biol.* 2012; Kim et al. *PNAS*, 2012).

Protein L is also a good model to study the pure foldase activity of TF on substrates under force. First, it has been thoroughly characterized by MT force spectroscopy in hours-long recordings at $\sim 4 - 50$ pN (Popa et al. *J Am Chem Soc.*, 2012) and it does not have any characterized misfolded state (Scalley, et. al. *Biochemistry*, 1997; Scalley, et. al. *PNAS*, 1997, Gu et al. *JMB*, 1997; Plaxco et al. *Nature Struct. Biol.*, 1999; Plaxco et al. *PNAS*, 1999). Even more importantly, protein L lacks of proline residues, which allows us to study purely the foldase activity without falling in any artifactual effect from proline isomerization, as previously described (Stoller et.al. *EMBO*, 1995). Also, as TF, protein L is a bacterial protein domain.

Additionally, it is important to note that we use an octamer of the 62 amino acids long protein L, which means a total length of ~ 496 amino acids (plus linkers). Furthermore, the interest of our study is to explore the functions of TF under force. And, under this scenario, polypeptide chains or proteins emerging from the ribosomal tunnel or translocating through molecular pores, which can then interact with TF, can have very different lengths. We believe that our magnetic-tweezers assay, with its stability and intrinsic force-clamp conditions, is able to mimic this scenario in a fairly good manner.

We know that substrate-bound TF could leave the ribosome while preserving its expanded state (see e.g. PMID: 17051157). How does the current study relates to this phenomenon?

The expanded state of TF is achieved when the chaperone docks onto the ribosome, and TF may remain expanded for some time after leaving the ribosome if it remains associated with the nascent chain (reviewer cites (Kaiser et al. *Nature*, 2006). However, it is known that the expanded state is not required for chaperone activity. Notably, TF is able to engage a variety of substrates and confer holdase/anti-aggregation activity in the absence of ribosomes (e.g. Saio et al. *Science*, 2014; Martinez-Hackert et al. *Cell*, 2009). Thus, the conformational expansion observed when TF docks onto the ribosome is not required for chaperone activity.

We are most likely observing interactions of TF in the compacted conformation with our unfolded substrates, which is also the case in other single molecule experiments conducted with TF (Mashaghi et al. *Nature*, 2013). Furthermore, the experiments in the reference provided by the reviewer are unable to discern if TF that is docked to ribosomes but still compacted can interact with nascent chains (see Kaiser et al. *Nature*, 2006, Figure 2A). Without this information, it is premature to speculate how the compacted versus expanded states might affect the foldase activity of TF either in the presence or absence of ribosomes.

(1) if TF acts as a foldase under force because of the entropic cost, we would expect this to be even more of a case in the absence of force. In the latter, the entropy cost would be larger and folding would be favoured subsequently.

Our data shows clearly that the interaction of TF with protein L substrates under force leads to three results: i) the folding probability increases; ii) refolding kinetics is accelerated; iii) unfolding kinetics remains unchanged. The combination of these three results within a conventional free energy landscape vision of protein folding imply the free energy of the unfolded state must *increase* relative to the folded state. This means that the unfolded state is destabilized and thus our discussion about the decrease in the entropy of the unfolded state. This scenario also fits with the literature regarding TF working as a cage or

cradle for polypeptide chains. A stretched polypeptide chain would accommodate in the TF groove, and as a consequence of this, see its conformational freedom reduced.

We agree with the reviewer that this effect should be even more prominent in the absence of force, because the restriction of this conformational freedom would be larger (being the entropy of the unfolded state larger). However, this argument might seem contradictory to prior works, which showed that TF works as a holdase in the absence of force (Kaiser et.al. Nature, 2006). Nevertheless, as pointed out before by the reviewer, the involved timescales might play a relevant role as the force is decreased. Above 6 pN, a single protein L domain folds in a timescale of ~ 10 s, competing thus with the binding-unbinding timescale of TF (which, although depends on how hydrophobic the substrate is, is being reported to have a binding timescale of $\sim 10^{-8}$ s, and an unbinding timescale of ~ 10 -35 s, so that the latter dominates, see Kaiser et.al. Nature, 2006). Interestingly, TF has the more prominent effect at these forces and higher, when folding is of the order or slower than the binding timescale. Protein L is a well-known folding model, being a good folder which folds in a fast timescale ($\sim 10^{-3}$ s), orders of magnitude faster than TF unbinding. Therefore, it is a plausible scenario that TF slows down folding, as its unbinding timescale is slower than that of protein L folding, working therefore as a holdase in the absence of force.

We thank the reviewer for this comment, which gave rise to a relevant discussion, which further improves our description of the TF activity. In order to further explore this issue, we designed a protocol to explore the influence of TF on protein L folding in the near-zero force regime. After a fingerprint pulse, we quench the force to a near-zero value of 0.7pN and allow the molecule to refold for different quench times of $t_{\text{quench}}=1, 2, 3, 5, 10, 15, 20, 30$ s, unfolding the molecule after to monitor the number of domains which folds during this quench time (see S3, panel A). We plot the fraction of folded domains as a function of the quench time in presence and absence of TF (see Fig. S3, panel B). Interestingly, we find how TF is not increasing folding at this very low force values. Indeed, our data suggests that TF is hindering folding. For example, after 30 seconds quench, about $\sim 100\%$ of the protein L domains are able to fold successfully, while in presence of TF, just about $\sim 60\%$ are recovered. This new data suggests that the mechanical force is actually modulating the activity of the chaperone. At low forces, when protein L is able to fold by itself within a fast timescale (with respect to the TF dynamics), TF slows down folding, working thus as a holdase. When the force is increased, slowing down folding to a timescale slower or comparable to that of TF binding-unbinding, TF works as a mechanical foldase, assisting folding by destabilizing the unfolded state.

We have included a new section on the SI detailing this new set of experiments (Figure S3). Also, this issue has been further addressed in the main text:

“This foldase activity on protein substrates under tension is independent of the known ability of TF to rescue the protein from kinetic traps (proline isomerization) or misfolded states, which has already been investigated with either bulk assays or non-equilibrium force rips [18, 31]. In order to further explore the activity of TF on protein L in the near-zero force regime, we design the protocol shown in Fig. S3, Panel A. After the fingerprint pulse, where the eight domains unfold, we quench to a near-zero force (~ 0.7 pN) for different times t_{quench} , unfolding subsequently to see how many domains folded during this time. We report the fraction of folded domains as a function of the quench time, in absence and presence of TF (Fig. S3, panel B). Interestingly, TF does not maintain the foldase activity in this near-zero force regime. Indeed, the chaperone seems to hinder the refolding transition since, after long quench times of 20 or 30 seconds, all protein L domains fold in absence of TF, while the presence of the chaperone decreases the fraction to $\sim 60\%$. This result is compatible with previous single molecule studies where TF seemed to stabilize the unfolded state or to protect partially folded states [16, 18].

This intriguing ability of the mechanical force to modulate the activity of TF can be described as a consequence of the competition between the timescales of protein L folding and TF binding-unbinding. TF binds to protein substrates

within times of $\sim 10^{-8}$ seconds [31], and the unbinding timescale is between 10 – 35 seconds, depending on the substrate [32]. This is much slower than folding of protein L in the absence of force, which is a good folder, folding within times of $\sim 10^{-2}$ - 10^{-3} seconds [33]. Therefore, it is plausible to think that binding of TF to an unfolded protein L, would indeed slow down the folding dynamics, as TF has to be expelled before protein folding. Nevertheless, the presence of the pulling force greatly slows down the folding dynamics, which is in a 10 – 100 second timescale in the 6 – 10 pN range (see Fig. 2B), comparable or even slower than that of TF unbinding. Therefore, upon TF binding, if this low entropy bound state is conducive to folding, and the free energy of folding is greater than the free energy of binding of the unstructured polypeptide to TF, the protein L domain will fold and expel TF within a faster timescale than that of protein L folding. This could be a possible reason that TF can work as foldase without ATP, unlike other chaperones such as HSP60 or HSP70 [11, 34]."

(2) The claim that HSP70 cannot bind unfolded chains without ATP cycle is incorrect.

We never mention that HSP70 cannot bind unfolded chains without ATP in our manuscript. To stress the foldase activity of TF in the *absence* of ATP we state *"TF can work as foldase without ATP, unlike other chaperones such as HSP60 or HSP70"*. This activity is different from other common chaperones such as HSP60 (GroEL) or HSP70. Although GroEL can bind to polypeptide chains, ATP is essential to initiate its foldase activity (Hartl et al Nature, 2011). Similar to GroEL, for the foldase activity of HSP70, ATP binding and hydrolysis are required both in vitro and in vivo and it has been stated *"Genetic and biochemical evidence clearly demonstrates that ATP hydrolysis is essential for the chaperone activity of all Hsp70 proteins tested so far"* (Mayer et al, Cell. Mol. Life Sci., 2005).

(3) There are many holdases among cellular chaperones and they are typically known to suppress folding. This might seem inconsistent with the model proposed here.

This question can be answered with the discussion on the timescales of folding and TF binding-unbinding, and our new experimental data on the holdase activity of TF in the near-zero force regime (see question (1) and Fig. S3).

(4) The author might want to consider interaction between TF and loosely folded protein L as an alternative model that could explain the results. The fact that the unfolding forces of the fully folded protein L did not rise in the presence of TF does not necessarily rule out a model that TF promotes folding by transiently protecting a partially folded state.

Previously, B. Bukau and S. Tans reported that TF binds and stabilized partially folded structures to promote correct folding (Mashagui, *et.al.* Nature, 2013). However, this work was done with Maltose Binding protein (MBP) a protein, which has more complex different folding landscape than protein L, which folds within a simple two-state scenario (Plaxco Nat. Struct. Biol. 1999).

It is important to note that, in our experiment, we studied the effect of TF chaperone on a substrate protein L under force. Thus the possibility of a loosely folded domain under force is rather unlikely, or at least, should be detected as different length steps in our step size distribution. Furthermore, Protein L has been largely characterized as a good folder, with no presence of misfolded states and a well-characterized single transition state (Popa et al. J Am Chem Soc, 2016; Scalley, et. al. Biochemistry, 1997; Scalley, et. al. PNAS, 1997; Gu et al. JMB, 1997; Plaxco et al Nature Struct. Biol., 1999; Plaxco et. al. PNAS, 1999). These results are in agreement with our previous work, stating that folding of protein L domains is "one to one", so no loosely folded domains appear to exist, at least within our experimental resolution (Popa, et.al. JACS 2017). To further clarify, we included a figure in the supplementary material (Fig. S4) showing that the step sizes of protein L in presence and absence of TF are identical, so TF does not induce any perceptible

conformational change on the folded or unfolded states of protein L. Therefore, given our experimental scenario, the depicted situation is quite unlikely. TF seems to bind to unfolded protein domains (stretched to a certain length imposed by the given force) and subsequently promote folding.

(5) It would be interesting to see how the authors put their findings in the context of protein translocation literature.

The arguments raised in the model depicted in Figure 4 are completely general for any polypeptide chain translocating through a molecular pore and folding at its mouth. In fact, we try to set it in this general perspective in the manuscript, although for the case of TF, the motivation appears for the case of the ribosomal tunnel. We discuss this in the context of reference [12] briefly in the introduction and discussion, setting the general scenario, applicable not only to TF but also to the translocon pore. In general, any translocating polymer, confined in a tunnel would be subject to a mechanical tension, as this argument comes from very general physical concepts, as stated in reference [34] and further discussed on [40]. Therefore, protein folding on the mouth of the tunnel would have to fight against this pulling force, which might hinder folding. TF is present in large excess in the cytoplasm (50 μM). Thus, we propose a new mechanism by which chaperones help protein folding against higher tensions. Hence, the term “mechanical foldase”.

(6) Given the proposed model, I am wondering why binding of TF to a chain would not introduce correlation into folding of neighbouring proteins. TF has a patchy interface that binds both hydrophobic and hydrophilic chains.

The length of a stretched single domain protein L at 10 pN is ~ 14 nm and the cavity size of TF is estimated to be ~ 10 -11 nm (Saio, et.al. Science, 2014). Thus, it is hard to think that TF could bind to more than a single domain or induce any correlation to neighboring domains. In the low force regime (4 – 10 pN), we observe, both in presence and absence of TF, single folding events, characterized by a measured step size following polymer physics model (see Popa, et.al. JACS, 2017). In this sense, no population of longer step sizes associated with simultaneous folding of domains is observed. This suggests that domain folding occurs one by one when assisted by TF. In order to further prove this, we have added a new section and figure (Fig. S4) in the SI showing the step size in presence and absence of TF, which are indeed identical, ruling out the possibility of correlated folding steps.

(7) I am surprised that the authors highlighted the changes to the folding landscape. I would not really call this “reshaping”. Nothing has changed “during” folding. And by the way, TF has been long known to bind chains, from both single molecule and bulk studies.

As it is clearly shown by our data, TF increases the folding probability of protein L under force and accelerates the refolding kinetics. The folding probability is a measure of the *equilibrium* relative population of the folded and unfolded states, therefore it is univocally related with the equilibrium free energy difference between the folded and unfolded state. Indeed, $\Delta G_{U-F} = -kT \log(P_f/(1-P_f))$, so an increase in P_f leads to a decrease in ΔG and thus a destabilization of the unfolded state relative to the folded. The acceleration of the refolding kinetics is a direct consequence of this, as the destabilization of the unfolded state will consequently decrease the folding free energy barrier. Thus, TF is changing the free energy difference between folded and unfolded and hence *reshaping* the folding free energy landscape. In order to further clarify this aspect, we have now included a supplementary figure (Fig. S2) showing the free energy difference between folded and unfolded state as a function of the pulling force in presence and absence of TF. TF decreases the free energy difference in the mentioned range of forces.

(8) In the entropic picture proposed above, the entropic terms associated with TF dynamics as well as the enthalpic terms associated with dimerization of TF have not been taken into consideration. See PMID:23565160.

One concern I have is that the major part of this study is based on 500 μM concentration of TF, i.e. the dimeric regime, while the whole study has been interpreted in the context of ribosome associated TF and translocation. Ribosome bound to TF is very different in terms of its conformation and available interacting surface with soluble monomeric TF and is obviously not comparable with dimeric TF. At

concentrations of a few hundred nM the authors did not see significant effects. How do the authors respond to this?

As discussed on the response to reviewer 1, question 3, our experimental data indicates that most of the effect of TF on the folding properties of protein L under force comes before the dimerization concentration (18 μ M). Therefore, monomeric TF seems to be able to shift by itself the folding probability under force. We include Fig. S5 to further clarify this point.

I found the theoretical study interesting for polymer physicist but I believe it is too simplistic for the purpose of this paper. The biological complexity of the system has not been taken into account. Additionally, the force applied in the experiment to the two ends of the protein L differs from the configuration used in the model.

We are very aware that many biological details have been omitted for the theoretical model of figure 4. Nevertheless, we believe that one of the strengths of the model is its the simplicity. With very simple physical arguments based on de Gennes's polymer theory (see *Scaling Concepts in Polymer Physics*, de Gennes, Cornell University Press) we are able to work out the optimal polymer length confined in a generic molecular tunnel and actually obtain a value which are very much in the order of what biologically found. We believe that the key arguments of the model are rather insensitive to the biological details of each particular system. Any confined polypeptide chain will be subject to an effective tension. Also, protein folding in the mouth of the tunnel will generate an increased tension due to the shortening of the polypeptide chain, so folding will have to "survive" against this increased force. Despite their generality, we believe these concepts have not been stated clearly in the field to date. Chaperones might play a relevant role by increasing the expected folding output.

Regarding the configuration issue, we do not believe it to be too different from the one employed in our experiments. The polypeptide chain is being synthesized on one end, while the other end is escaping through the mouth of the tunnel. Upon protein folding, this shortening generates a force towards the tunnel (dial shown in Fig. 4). Subsequently, the steric exclusion in the mouth of the tunnel generates a "normal force" in the opposite direction. Effectively, this is equivalent to a tension that runs through the polypeptide chain; analogous to the force we apply in our experiment. The key point is that folding occurs against a mechanical tension, given significance to our discovery of TF working as a mechanical foldase.

Minor

Page 8, Line 2: Please correct for spelling error in "though".

We thank reviewer 2 for the correction. We have corrected it in the manuscript.

Reviewer #3 (Remarks to the Author):

The present study aims at elucidating the mechanism of action of the molecular chaperon TF (Trigger Factor). The authors use magnetic tweezers to monitor folding of protein L under mechanical force in the presence of different concentrations of TF. The results of these studies suggest that TF facilitates the folding of protein L in the physiological force range of 4-10 pN by reducing the conformational entropy of the unfolded state. Moreover, the results show that the foldase activity of TF is force dependent as the effective affinity of TF for protein L decreases at higher forces. I found this work original, and the reported data elegant and carefully analyzed, supporting the major conclusions drawn from them. The manuscript is generally well written and clear. Consequently, I recommend this work for publication in Nature Communications, after revision to address the following minor points

1. On page 8 the authors write “Notably, the ribosomal exit tunnel can accommodate 35 amino acids inside the tunnel[33]”. In reality, reference 33 suggests 30 amino acids in the ribosomal tunnel. Thus “35 amino acids” should be changed into “30 amino acids”.

We acknowledge the reviewer for the remark. We have corrected the number in the manuscript.

2. At the end of page 8, the authors make a list of “several uncounted factors” that can affect the “actual force-to-extension relation”. I suggest quoting at least one reference for each suggested factor to support their claims.

We thank the reviewer for the suggestion. References [39] and [40] has been added to the main text.

Reviewers' comments:

Reviewer #1 (Remarks to the Author):

In the previous review, this reviewer most importantly asked about the effect of the E. coli trigger factor (TF) on the folding energy landscape of protein L domains. This is because although the authors mainly used the folding probability in their analysis, the current study can be more directly compared with the previous literature and other single-molecule force spectroscopy studies when one knows the resultant changes in the free energy landscape. In the revised manuscript, the authors now present the ΔG values (the free energy difference between the folded and the unfolded states) as a function of force in Supplementary Figure 2.

The main experimental data in the manuscript repeatedly show that the effect of TF is most evident in the force range between 7 and 9 pN (e.g., Figure 2B and C, Figure 3 and Figure. S5). The calculation of the ΔG value show that there is less than 3 kBT difference in the corresponding force range. At 9 pN, there is only 1 kBT difference which would be easily masked by ambient thermal fluctuations.

Given this ΔG value calculation, a more realistic picture is likely that in the presence of TF, the protein L domain might spend an increased fraction of time in the folded state (thus increase in the folding probability), but this TF does not induce a stably folded structure of the protein L in the force range between 7 and 9 pN. When the force becomes smaller than 6 pN, the mechanical tension is not high enough to perturb the folding process (e.g., Fig. 3, black square symbols) and the protein L can show very fast folding even in the absence of TF (e.g., Fig. 2B). In fact, the authors added new claims that in the low force regime close to zero tension, the TF actually works as a holdase.

Therefore, the effect of TF observed in this work as a foldase might be too weak (not being able to induce stable folding) and limited to a small force range to play a significant role in a physiological systems.

This reviewer finds that other major comments (Major comments 2 to 4) have been well addressed by the authors.

Minor comments.

This reviewer agrees with the approach the authors took to derive the ΔG values out of the folding probability. Technically, however, this graph does not contain the experimental data points that must have been used for the authors to plot the dependence of the ΔG values on the mechanical tension. The dependence seems to be highly linear suggesting that the energy landscape follows a simple picture suggested by the Bell equation. This reviewer wonders whether this is indeed true because the distance between the folded and unfolded states along the reaction coordinate is relatively long, amounting to approximately 7 nm (6 pN increase causes 10 kBT difference, that is, 40 pN times nm at room temperature). When the distance is long, the simple Bell equation usually requires revision because the states move in the assumed reaction coordinate. The experimental data points used for curve fitting must be plotted.

Reviewer #2 (Remarks to the Author):

I thank the authors for carefully considering the comments/suggestions and for their efforts to improve the manuscript. I am of the opinion that this is an interesting study and can be published in a decent journal (e.g. Scientific Reports) in its current form. I am not fully convinced that Nature Communications would be the best home for this content, given the limitations of the study (e.g. choice of the substrate and simplicity of the model) and the broadness of the claimed

relevance (foldase activity of ribosome bound-TF during translation and free TF-catalyzed refolding during mechanical processes in (muscle) cells).

Minor:

Regarding force and TF, the authors may want to note a recent paper by von Heijne et al. (PMID: 26906929).

REVIEWER #1

In the previous review, this reviewer most importantly asked about the effect of the E. coli trigger factor (TF) on the folding energy landscape of protein L domains. This is because although the authors mainly used the folding probability in their analysis, the current study can be more directly compared with the previous literature and other single-molecule force spectroscopy studies when one knows the resultant changes in the free energy landscape. In the revised manuscript, the authors now present the ΔG values (the free energy difference between the folded and the unfolded states) as a function of force in Supplementary Figure 2.

The main experimental data in the manuscript repeatedly show that the effect of TF is most evident in the force range between 7 and 9 pN (e.g., Figure 2B and C, Figure 3 and Figure. S5). The calculation of the ΔG value show that there is less than 3 kBT difference in the corresponding force range. At 9 pN, there is only 1 kBT difference which would be easily masked by ambient thermal fluctuations.

Given this ΔG value calculation, a more realistic picture is likely that in the presence of TF, the protein L domain might spend an increased fraction of time in the folded state (thus increase in the folding probability), but this TF does not induce a stably folded structure of the protein L in the force range between 7 and 9 pN. When the force becomes smaller than 6 pN, the mechanical tension is not high enough to perturb the folding process (e.g., Fig. 3, black square symbols) and the protein L can show very fast folding even in the absence of TF (e.g., Fig. 2B). In fact, the authors added new claims that in the low force regime close to zero tension, the TF actually works as a holdase.

Therefore, the effect of TF observed in this work as a foldase might be too weak (not being able to induce stable folding) and limited to a small force range to play a significant role in a physiological systems.

This reviewer finds that other major comments (Major comments 2 to 4) have been well addressed by the authors.

Reviewer #1 main concerns come from the free energy interpretation of the folding probability, reported in Fig. S2. However, we believe this representation has led to confusion, shifting the focus from the actual findings of our work. Our manuscript presents the first measurement of the folding probability of a protein under force in presence of a molecular chaperone. Our data unambiguously shows that the TF increases the equilibrium population of the folded state and greatly accelerates the folding kinetics, precisely on the physiological relevant force range.

Following Reviewer #1 suggestion, we translated the folding probability to a free energy representation by simply using Boltzmann-Gibbs distribution. Therefore, it is important to note that Fig. S2 does not add any new information to our findings, and also, that it is representing an equilibrium magnitude. However, this plot can easily be misinterpreted when trying to compare it with bulk or ensemble measurements. Reviewer #1 is mainly concerned that the foldase activity of TF is “too weak”. However, our data clearly shows the opposite, since the folding probability is increased up to 40% at 8 pN (this means that the population of the folded state is doubled at this force), and the folding kinetics are accelerated by a factor of 10. Even at 8.9 pN (where the estimated free energy change is ~ 1 kT), the population of the folded state rises from $\sim 25\%$ to $\sim 60\%$, which is nearly an increase of a factor of 3, certainly not something to be masked by thermal fluctuations, which indeed have no effect on equilibrium magnitudes. This factor of 3 increase in the presence of the chaperone comes from sampling more than two hours of folding trajectories in aggregate between all of the experiments at 8.9 pN, both with and without the chaperone. We have equally as much data for the other forces represented in the folding probability.

Therefore, we believe that the picture suggested by the reviewer is inaccurate. TF does not seem to change the stability of the folded state of protein L as the unfolding rates are unchanged (Fig. S1). Rather, TF decreases the stability of the unfolded state, (increase in folding probability) and also accelerates the refolding kinetics. No matter if protein L already folds in the absence of TF, the kinetics are accelerated even more in the presence of TF (Figure 2B).

Regarding the force range, TF does not work on a limited force range, but precisely on the relevant force range in physiology. Proteins fold under force generally up to 10 pN, and it is the between 6 and 10 pN, where the probability of folding decreases abruptly, so mechanical foldases might have a relevant role by increasing the folding output. In particular, nascent protein chains has been reported to fold optimally between 7 and 10 pN (Goldman et.al. Science 2015), and the force produced by protein folding helps rescuing stalled chains in the ribosomal tunnel. Therefore, we believe our discovery can be very relevant in this context, since the activity of TF takes place just on the physiological force range, where it can maximize the force generated by protein folding, as we predict in Fig. 4C.

For the reasons exposed above, the mechanical foldase activity of TF we report is clear and sufficiently relevant. Although we followed the suggestion of Reviewer #1, we have removed Figure S2 because: (a) the discussions about the free energy changes induced by the chaperone are misleading if interpreted with a “bulk” thermodynamic perspective, which studies different folding pathways than single molecule force spectroscopy; and (b) the free energy plot detracts from our findings of the large shift in the folded populations in the presence of chaperone, as depicted throughout the manuscript (Fig. 2C, Fig. 3).

Minor comments.

This reviewer agrees with the approach the authors took to derive the ΔG values out of the folding probability. Technically, however, this graph does not contain the experimental data points that must have been used for the authors to plot the dependence of the ΔG values on the mechanical tension. The dependence seems to be highly linear suggesting that the energy landscape follows a simple picture suggested by the Bell equation. This reviewer wonders whether this is indeed true because the distance between the folded and unfolded states along the reaction coordinate is relatively long, amounting to approximately 7 nm (6 pN increase causes 10 kBT difference, that is, 40 pN times nm at room temperature). When the distance is long, the simple Bell equation usually requires revision because the states move in the assumed reaction coordinate. The experimental data points used for curve fitting must be plotted.

Figure S2 was calculated using the sigmoidal fits to the folding probability. These fits are not based on any particular model, and therefore should not be used to raise any conclusion on the physics of the system. The exponential character of the sigmoidal function leads to the apparent linear behavior of the free energy, since it is just a logarithmic representation of the folding probability. We are very aware that the distance between the folded and unfolded states follows common polymer models such as the freely-jointed chain. We report this clearly in the step sizes in Fig. S3. We do not think the apparent linear behavior is due to any Bell-like picture, since this is the representation of an equilibrium magnitude and not a free energy barrier, and therefore they are not related. Rather, this is due to the simple sigmoid utilized to fit the folding probabilities.

Reviewer #2 (Remarks to the Author):

I thank the authors for carefully considering the comments/suggestions and for their efforts to improve the manuscript. I am of the opinion that this is an interesting study and can be published in a decent journal (e.g. Scientific Reports) in its current form. I am not fully convinced that Nature Communications would be the best home for this content, given the limitations of the study (e.g. choice of the substrate and simplicity of the model) and the broadness of the claimed relevance (foldase activity of ribosome bound-TF during translation and free TF-catalyzed refolding during mechanical processes in (muscle) cells).

We thank Reviewer #2 for accepting that we addressed correctly all of his comments and suggestions regarding

our manuscript. We believe it helped improving our work. However, we do not understand his new position, specially regarding the lack of a clear scientific argument which could be subsequently addressed.

Reviewer #2 claims that our study is limited given the choice of our substrate and the limitations of the model. We strongly disagree with this position, since indeed protein L was carefully chosen as an adequate and relevant substrate for our study. Also, we put the activity of TF in a plausible physiological context, proposing a model which, despite the claimed simplicity, is able to produce quantitative information which agrees with the numbers found in the literature, as we discuss in the manuscript.

Protein L is a standard protein for studying protein folding, and it has been used extensively in numerous works by different groups and scientists (Scalley, et. al. *Biochemistry*, 1997; Scalley, et. al. *PNAS* 1996; Gu et al. *JMB*, 1997; Plaxco et. al. *Nature Struct. Biol.*, 1999; Plaxco et. al. *PNAS*, 1999; Maity et. al. *J. Am. Chem. Soc.* 2016; Tae Yeon et. al. *J. Mol. Biol.* 2012; Kim et al. *PNAS*, 2012), which has also well characterized using magnetic tweezers force spectroscopy (Popa et al. *J Am Chem Soc.*, 2016, Valle Orero et al. *Angew. Chem. Int. Ed.* 2017, Liu et al. *Biophysical Journal* 2009). Importantly, protein L lacks of proline residues, so it allows us to study a pure foldase activity, not product of an artifact of the proline isomerization activity of TF (Stoller, et.al. *EMBO* 1995). Finally, both protein L and TF are bacterial proteins, so it is a perfectly relevant substrate since working with bacterial protein substrates to study bacterial chaperones is universal in the field (Gupta et al 2014 *JMB*, Haldar et al 2015 *JMB*, Tang et al *Cell* 2006, Chakraborty et al, *Cell* 2010, Georgescauld et al, *Cell* 2014, Sharma et al, *Cell*, 2008, Mashaghi et al, *Nature*, 2013, Siao et/ al *Science* 2014).

Regarding the claimed relevance, force is being recognized increasingly as a key component of protein folding in many physiological situations (Goldman et.al. *Science* 2015, Rivas-Pardo et.al., *Cell Reports* 2016, Aubin-Tam, *Cell* 2011, Ismail N. *Nat Struct Mol Biol* 2012). We never claim that TF can have a role in mechanical processes in muscles, since basically it is not present there. However, muscle mechanics are perhaps the most classical example of protein folding in presence of force, since titin domains fold and unfold in the force regime at which the sarcomere operates, assisting muscle contraction (Rivas-Pardo et.al., *Cell Reports* 2016). The presence of molecular chaperones with a similar role to TF (allowing folding to occur against higher forces) would certainly have a relevant physiological role, since it would allow maximizing the work done by protein folding. In this sense, we claim our findings might be of relevance since protein folding under force is ubiquitous in cells, and therefore, mechanical chaperones can certainly have a relevant role in this context.

Minor:

Regarding force and TF, the authors may want to note a recent paper by von Heijne et al. (PMID: 26906929).

We were aware of this reference which we indeed cite in our manuscript (reference 2).

REVIEWERS' COMMENTS:

Reviewer #3 (Remarks to the Author):

For the first time, the present study reports on the folding probability of a protein (protein L) under mechanical force in the presence of a molecular chaperone TF (Trigger Factor).

The first major concern raised by reviewer 1 is related to the low ΔG (a few KBT) between the folded and unfolded states of protein L measured by the authors in the force range 7-9 pN. According to reviewer 1 this low ΔG values does not fully support the conclusion drawn by the authors that chaperon TF favours the formation of a stably folded state of protein L; in fact according to reviewer 1 these low ΔG values are too close to ambient thermal energy to be reliable. The authors replied that the increase in the population of protein L folded states in the presence of chaperone TF that they measured in their experiments (from ~20% to ~60%, Figure 3) is large enough to exclude any possible doubt on their findings. Similarly, they pointed out that the folding probability increase of 40% at 8 pN and the escalation in the folding kinetics by a factor of 10 that they show in Figure 2 are significant enough to strongly support a mechanical foldase activity for chaperone TF.

The second concern raised by reviewer 1 is related to the force range (7-9 pN) in which the measurements were carried out. According to reviewer 1 this forces are too low to be physiologically relevant. The authors disagree with this concern pointing out the fact that according to the current literature nascent protein chains fold in the force range between ~7 and ~10 pN.

In general I think that the authors have addressed correctly all the concerns raised by reviewer 1, and I believe that the quality of the data and the novelty of the experimental approach presented by Shubhasis Haldar et al. in this paper justify a publication in Nature Communications.

REVIEWERS' COMMENTS:

Reviewer #3 (Remarks to the Author):

For the first time, the present study reports on the folding probability of a protein (protein L) under mechanical force in the presence of a molecular chaperone TF (Trigger Factor).

The first major concern raised by reviewer 1 is related to the low ΔG (a few KBT) between the folded and unfolded states of protein L measured by the authors in the force range 7-9 pN. According to reviewer 1 this low ΔG values does not fully support the conclusion drawn by the authors that chaperon TF favours the formation of a stably folded state of protein L; in fact according to reviewer 1 these low ΔG values are too close to ambient thermal energy to be reliable. The authors replied that the increase in the population of protein L folded states in the presence of chaperone TF that they measured in their experiments (from ~20% to ~60%, Figure 3) is large enough to exclude any possible doubt on their findings. Similarly, they pointed out that the folding probability increase of 40% at 8 pN and the escalation in the folding kinetics by a factor of 10 that they show in Figure 2 are significant enough to strongly support a mechanical foldase activity for chaperone TF.

The second concern raised by reviewer 1 is related to the force range (7-9 pN) in which the measurements were carried out. According to reviewer 1 this forces are too low to be physiologically relevant. The authors disagree with this concern pointing out the fact that according to the current literature nascent protein chains fold in the force range between ~7 and ~10 pN.

In general I think that the authors have addressed correctly all the concerns raised by reviewer 1, and I believe that the quality of the data and the novelty of the experimental approach presented by Shubhasis Haldar et al. in this paper justify a publication in Nature Communications.

We thank reviewer 3 for his comments.